# T cells in the brain enhance neonatal mortality during peripheral LCMV infection

**Laurie L. Kenney, Erik P. Carter⬤, Anna Gil, Liisa K. Selin⬤***

University of Massachusetts Medical School, Department of Pathology, Worcester, Massachusetts, United States of America

* Liisa.Selin@umassmed.edu

**Data Availability Statement:** All relevant data are within the manuscript.

**Funding:** This work was funded by National Institutes of Health: AI-46578 (LKS), AI-46629 (LKS), AI-109858 (LKS), training grant T32-AI-

## Abstract

In adult mice the severity of disease from viral infections is determined by the balance between the efficiency of the immune response and the magnitude of viral load. Here, the impact of this dynamic is examined in neonates. Newborns are highly susceptible to infections due to poor innate responses, lower numbers of T cells and Th2-prone immune responses. Eighty-percent of 7-day old mice, immunologically equivalent to human neonates, succumbed to extremely low doses (5 PFU) of the essentially non-lethal lymphocytic choriomeningitis virus (LCMV-Armstrong) given intraperitoneally. This increased lethality was determined to be dependent upon poor early viral control, as well as, T cells and perforin as assessed in knockout mice. By day 3, these neonatal mice had 400-fold higher viral loads as compared to adults receiving a 10,000-fold ($5 \times 10^4$ PFU) higher dose of LCMV. The high viral load in combination with the subsequent immunological defect of partial CD8 T cell clonal exhaustion in the periphery led to viral entry and replication in the brain. Within the brain, CD8 T cells were protected from exhaustion, and thus were able to mediate lethal immunopathology. To further delineate the role of early viral control, neonatal mice were infected with Pichinde virus, a less virulent arenavirus, or LCMV was given to pups of LCMV-immune mothers. In both cases, peak viral load was at least 29-fold lower, leading to functional CD8 T cell responses and 100% survival.

## Author summary

As in adults the general principle that the balance between viral load and immune responses determines disease outcome applies in neonates, although the immune environments and exact mechanisms differ. A better understanding of these differences will improve strategies to optimize protection of the highly susceptible neonatal population. These results also suggest that the environment of the brain may protect T cells from exhaustion.

007349-16 (LLK). The funders had no role in study design, data collection and analysis, decision to publish, or preparation of the manuscript.

**Competing interests:** The authors have declared that no competing interests exist.

## Introduction

Newborns and infants are highly susceptible to infections. A clear understanding of how young developing immune systems respond to viral infections and immunizations and generate memory responses is necessary to optimize protection of this population. We questioned how the developing immune system of a 7-day-old neonatal mouse, as a model to study human newborns, would respond to infection, clear virus and form memory compared with an adult mouse.

There are several differences in the neonate that cause altered immune responses that cannot be predicted from studying the established adult immune system. Differences in the size, breadth, affinity and specificity of immune responses have been reported in neonates, with disparities in the innate, cellular and humoral arms of the immune system [1–4]. The developing immune system of the neonate has previously been inaccurately described as "immature". However, a new perspective has emerged that the neonatal immune system is functional, but highly plastic as it is undergoing tolerization to self and microbiota, while also being able to fight off infections [4, 5]. Neonates do not have their own immunological memory, which is the main protective mechanism for adults from re-exposure to a pathogen. Additionally, neonates have very low numbers of immune cells. Both T and B cell frequencies are reduced 15-30-fold in the peripheral blood of humans and spleens of mice [6]. Functionally, the immune cells in neonates differ from adults. Both intrinsic and extrinsic differences within the T cell response to infections have been identified in both young mice and humans [2, 4, 7–9]. Under the right situations neonates have been found to be capable of producing Th1/cytotoxic T cell responses [10–12]. However, it is still not completely understood why this happens during some infections, but not others. For example, high doses of murine leukemia virus induced a Th2 response in neonatal mice, while low doses allowed for the induction of adult-like cytotoxic CD8 T cell and Th1-skewed CD4 T cell responses [12]. Young mice infected or immunized with strong Th1/CTL inducing agents, such as DNA vaccines or UV-killed viruses, can produce adult-like Th1/CTL responses. The overall view on neonatal immune responses is that low, persistent levels of antigen can promote CTL/Th1 responses, while higher doses induce Th2 responses or tolerance, which can be induced by clonal exhaustion of the immune response. We questioned how CD8 T cell responses would develop in neonatal mice after infection with a fast replicating, strong CTL inducing virus, such as LCMV, known to induce clonal exhaustion at high doses [13, 14].

LCMV infection of adult B6 mice is a well characterized model of CD8 T cell immunity and immunopathology. However, the induction of a T cell response in young mice after LCMV infection has not been well studied. When adult B6 mice are infected with the Armstrong strain of LCMV intraperitoneally (ip) virus is cleared from all organs by day 8, and the CD8 T cell response peaks between day 8 and 9, followed by the peak of the CD4 T cell response between day 9 and 11 [14–16]. Clearance of primary LCMV virus infection in adults is dependent on cytotoxic T cell responses. LCMV Armstrong infections in adult mice result in a fully functional and efficient T cell response that clears virus in approximately one week. Alternatively, the clone 13 strain of LCMV induces a wide range of disease severity and immunological response depending on the dose administered [17–19]. LCMV clone 13 is a clonal strain isolated from a mouse persistently infected with LCMV Armstrong [13] and has a single amino acid distinct from the Armstrong strain [20, 21]. This mutation alters the affinity of the LCMV clone 13 for the α-distroglycan receptor allowing it to rapidly systemically infect mice [20, 21]. At a low dose infection ($2x10^4$ PFU) with LCMV clone 13, the kinetics of viral replication and the immune response are similar to LCMV Armstrong. When LCMV clone 13 is administered at a 100-fold higher dose ($2x10^6$ PFU) rapid viral replication to high viral titers

drives a complete clonal exhaustion of the T cell response. During clonal exhaustion, repeated exposure to high doses of antigen or lack of CD4 T cell help drives a loss of function in a step-wise manner. The loss of cytokine production occurs in the order of IL-2, then TNF and then IFNγ, and clonal exhaustion finally drives apoptosis of antigen-specific T cells [14]. Cytotoxic function is also lost early approximately at the same time as IL-2 production is lost. At an intermediate dose of LCMV clone 13 (2x10$^5$ PFU) viral loads are only high enough to drive a partial clonal exhaustion of the LCMV-specific response with reduced cytotoxic function and decreased IFNγ production. This results in sub-optimally functioning CD8 T cells that allows for virus to persist and the remaining functional T cells mediate the induction of severe immunopathology leading to death [17–19]. The balance of the viral load and the ability of the T cell response to clear virus determine disease severity after LCMV infection.

The age of infection with LCMV Armstrong also influences disease outcome through skewing the balance between viral titers and ability of the immune system to clear virus. Newborn mice infected with LCMV undergo central tolerance or clonal exhaustion, leading to persistent life-long infection [22, 23]. Virus replicates quickly systemically, including in the thymus, and results in rapid deletion of LCMV-specific T cells with no immunopathology and high systemic viral load. Only one study has been done to examine T cell responses to LCMV infection in older neonates [24]. 14-day-old Balb/c infants cleared the WE strain of LCMV with delayed kinetics and little immunopathology [24]. These data would suggest that by 2 weeks of age the immune systems of infant mice have the functional capability to efficiently clear virus prior to the development of significant immunopathology or clonal exhaustion. We utilized the 7-day-old neonatal mice as a model for human newborns to study efficiency of CD8 T cells in response to LCMV infection and the resulting immunopathology. At birth, mice are immunologically less mature than human newborns and at a greater risk of tolerance induction. Studies suggest that the 7-day-old mouse immune system is more similar to human newborns, including BCG-specific responses and naive T cell receptor (TCR) repertoires [4]. Hereafter, the 7-day-old mouse will be referred to as a neonate.

Here, we show that the neonate is highly susceptible to T cell-mediated lethality after LCMV Armstrong infection. Virus replicated to high titers soon after infection, driving a partial clonal exhaustion of the CD8 T cell response in the spleen. Mortality was mediated by perforin, most likely released from CD8 T cells. Replicating virus and more functional CD8 T cells were isolated from the brain. The protection of the CD8 T cells from clonal exhaustion within the brain may also be a major mediator of lethality. To further study the neonatal immune system, we used two alternative infection models that had reduced viral load. Passive immunity from LCMV-immune mothers and infection with a less virulent arenavirus, Pichinde virus, resulted in lower viral loads and the development of a fully functional adult-like CD8 T cell response and 100% survival. This study suggests that the balance between the efficiency of the immune response to clear a pathogen and the viral load determines the disease outcome in neonates.

## Results

### Day 7 neonates are highly susceptible to LCMV-mediated death

One-week-old uninfected mice had reduced frequencies and total numbers of CD4 and CD8 T cells as compared to 6-week-old adults (Fig 1A–1D). The CD8 T cell population consisted of only 1% of the spleen at 1 week of age and increased to 13% by week 6 of age, when mice are considered to be adults (Fig 1A). By total number there was 97% fewer CD8 T cells in spleens of neonatal mice (Fig 1B). CD4 T cell frequencies were on average 1.8% in 1 week old neonates

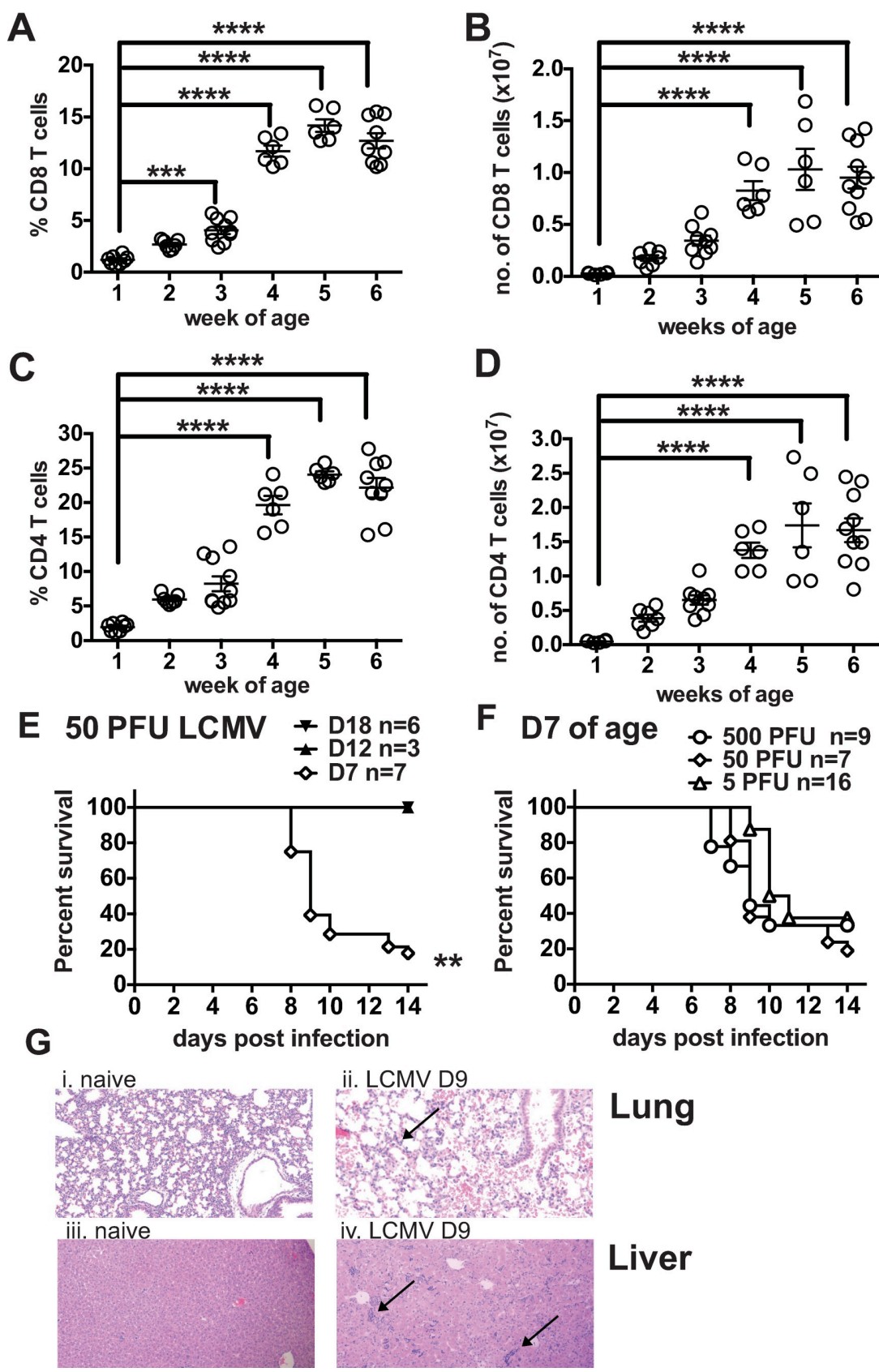

**Fig 1. 7-day-old neonatal mice are highly susceptible to LCMV(Armstrong)-induced mortality and immunopathology.** At week(s) 1, 2, 3, 4, 5 and 6 of age the frequency (A and C) and total number (B and D) of splenic CD8 (A and B) and CD4 (C and D) T cells were determined by surface staining from naïve mice. Data are from at least 2 similar experiments with 6–10 mice/group. (E) Mice at 18, 12 and 7 days of age were infected with 50PFU of LCMV Armstrong ip in 50ul and survival was monitored until day 14 post infection. (F) Day 7 neonates were infected with 500, 50 or 5 PFU of LCMV Armstrong and survival was monitored until day 14 post infection. Data are from 1–3 similar experiments. n = 3–16 mice/group. (G) Lung and liver sections were stained with H&E. (Gi) Naïve lung showed no pathology. (Gii) Day 9 post infection with LCMV showed mononuclear infiltrate and edema in the lung. (Giii) Naïve liver shows no pathology. (Giv) Liver from day 9 post LCMV Armstrong infection shows numerous patches of mononuclear infiltrates. Arrows highlight pathology.

and expanded 12-fold by week 6 of age (Fig 1C), consistent with previous findings [25]. Due to this significant reduction in T cells and the importance of these cells in clearing LCMV, we questioned how this would impact immunopathology, viral clearance and T cell immunity. Adult mice infected with the typically used dose of LCMV Armstrong ($5x10^4$ PFU) clear virus within one week, develop a strong CD8 T cell response, and have minimal immunopathology [14]. After preliminary studies, where neonates infected with adult doses of LCMV showed severe lethality, neonates were infected with a 1000-fold lower dose, 50 PFU, and still had over 80% mortality, with neonates succumbing to infection between 8–10 days post infection (Fig 1E). In contrast, when 12- or 18-day old mice were infected with 50 PFU of LCMV there was 100% survival, indicating that only mice younger than 12 days of age were exceptionally susceptible to LCMV-induced mortality (Fig 1E).

Neonates infected with 5, 50 and 500 PFU of LCMV all had similar mortality rates (62–80%), with similar kinetics (mean survival 7–10.5 days post infection) (Fig 1F). Histology of lungs and livers in surviving infected neonates displayed enhanced mononuclear infiltrates at day 9 post infection, which is the peak of the CD8 T cell response in adult LCMV infection and the time point when the majority of the neonates were succumbing to infection (Fig 1G). This high rate of mortality was surprising, as LCMV is a nonlytic virus and this strain of LCMV does not cause mortality in adult mice inoculated by this route. With other strains of LCMV, such as clone 13 when given at an intermediate dose ($2X10^5$ PFU), or if LCMV Armstrong is injected intracranially, T cell-mediated mortality has been found in adult mice [17–19, 26, 27].

## Surviving neonatal mice clear virus with delayed kinetics

In adult mice infected with LCMV Armstrong, virus replicates predominantly in the spleen and lymph nodes, but it does infect other organs including the liver and kidney [14, 28] (Fig 2A–2C). In this study, in adult mice LCMV replicated at low levels in the kidneys, lungs and liver, with viral loads peaking between 3.2–3.6 $\log_{10}$ PFU/gram of organ (Fig 2A–2C). The peak of virus replication was day 3 in the lung and day 6 in the kidneys and liver, with clearance from all organs on day 9 (Fig 2A–2C). Compared to adult mice, neonates had at least 400-fold higher viral loads, and viral clearance was delayed by 5 days in the liver and by up to 12 days in the kidney and lung. LCMV replicated in kidneys, lungs and liver to over 6 $\log_{10}$ PFU/gram of organ in neonates (Fig 3A–3C). Virus replicated to these high levels in every neonate tested, but, due to the high mortality rate, only a small surviving percentage could be examined at day 14 post infection. At this later time point the majority of the surviving mice had cleared virus and by 3 weeks post infection virus was not detectable (Fig 2A–2C). These data show that viral clearance was protracted in neonatal mice, even during infection with 10,000-fold lower doses of LCMV Armstrong (5 PFU). Furthermore, the levels of virus were significantly higher in neonates compared to adults, even though they are infected with a much lower dose.

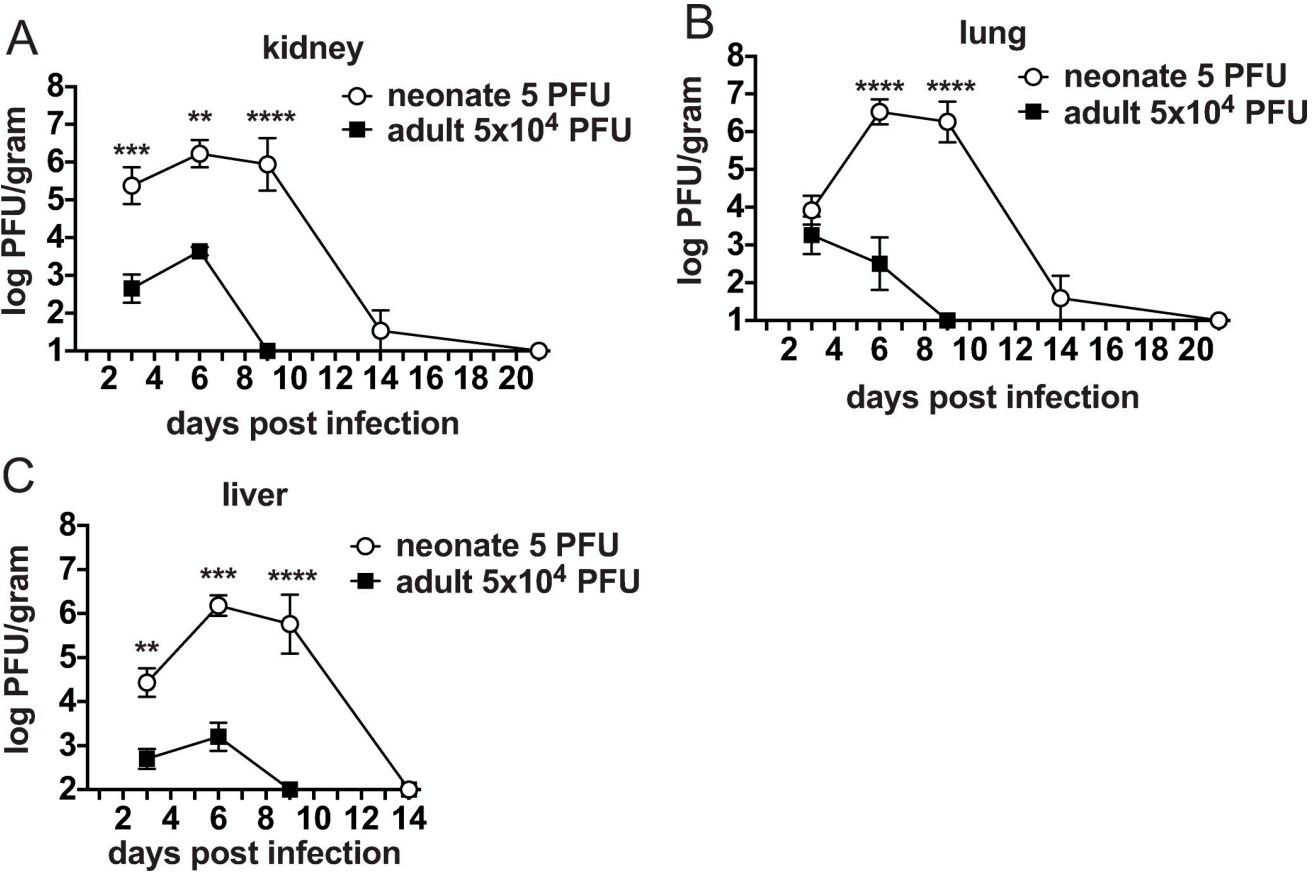

**Fig 2. Clearance of LCMV Armstrong is delayed in neonates.** A-C. Viral load was determined in (A) kidney, (B) lung and (C) liver of both neonates infected with 5 PFU and adults infected with $5 \times 10^4$ PFU of LCMV. Viral load was determined by plaque assay and $\log_{10}$ PFU was calculated per gram of organ to compare adult to neonate. Averages of 4–8 mice/group are shown from at least two separate experiments. The detection limit of the plaque assay for kidney, lung and brain is 1 $\log_{10}$; the liver is 2 $\log_{10}$.

## Neonatal mortality during LCMV infection is T cell and perforin-dependent

The death rate of neonates infected with low doses of LCMV Armstrong and the immunopathology of their vital organs resembled what has previously been observed in adult male mice infected with an intermediate dose ($2 \times 10^5$ PFU) of the more widely disseminating LCMV clone 13 strain administered intravenously (i.v.) [19]. In adult B6 mice an intermediate dose of LCMV clone 13 i.v. results in approximately 60% mortality, which was found to be mediated by T cell responses, as infection of TCRβ knockout (KO) mice and mice depleted of CD8 T cells resulted in 100% survival [17–19]. We questioned if neonates were dying of a similar mechanism. TCRβ KO neonates infected with 50 PFU of LCMV Armstrong had 100% survival, while wildtype B6 neonates had only ~20% survival (Fig 3A). To further examine which T cell subsets were causing death, neonates were depleted of either CD4 or CD8 T cells using monoclonal antibodies (Fig 3B). Neonates depleted of CD4 T cells showed no difference in either the kinetics or the frequency of mortality. Depletion of CD8 T cells significantly reduced mortality to only 55% and increased mean survival from 9.5 days in control neonates to 13 days. Furthermore, lethality in neonates was not directly due to viral load. TCRβ KO neonates at day 9 post infection had 6 $\log_{10}$ PFU virus/gram in kidney, lung and liver, levels that were

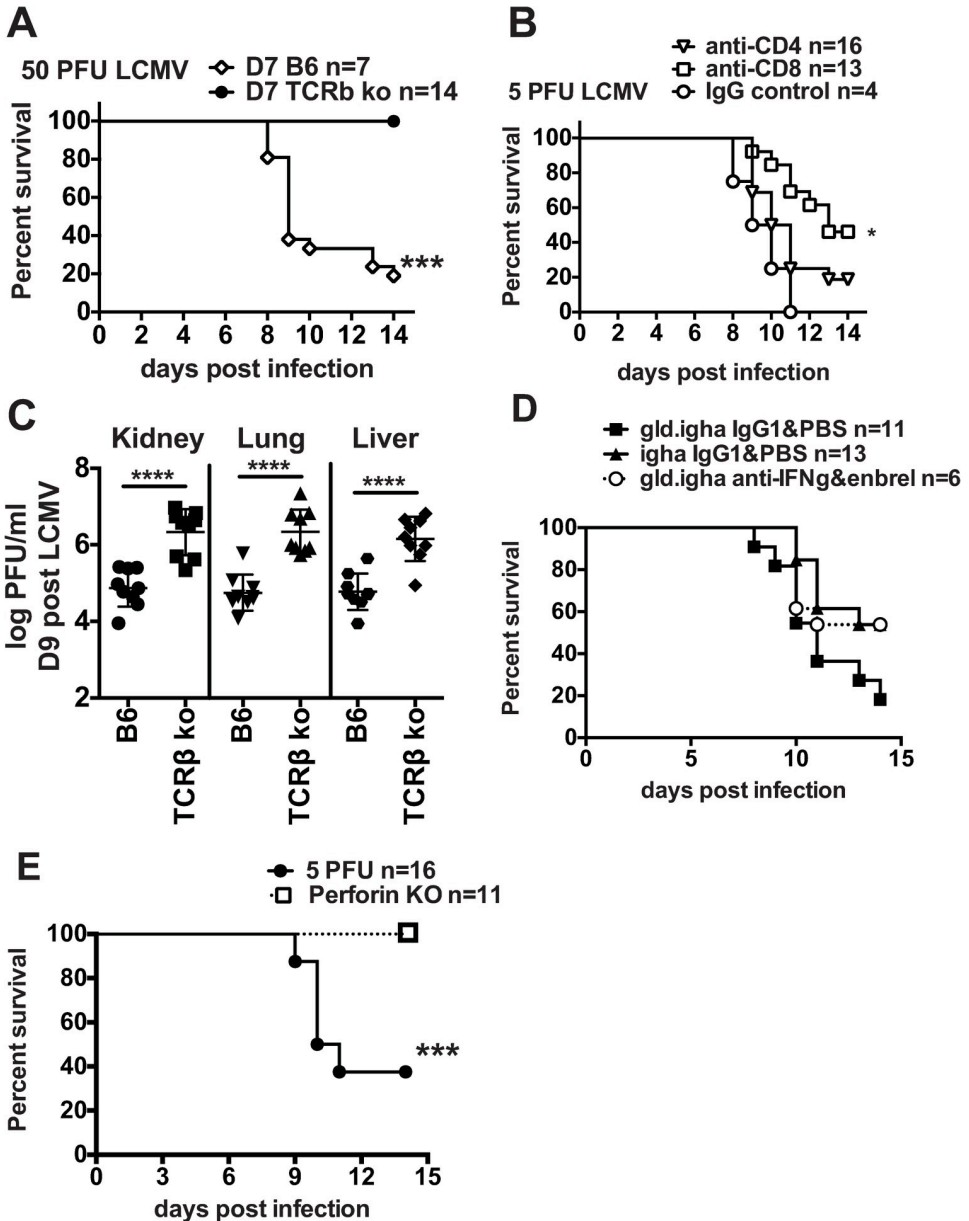

**Fig 3. 7-day-old neonates die from T cell-mediated immunopathology during LCMV Armstrong infection.** (A) 7-day-old TCRβ KO neonates (n = 14) and wildtype B6 controls (n = 7) were infected with 50 PFU of LCMV Armstrong and survival was monitored until day 14 post infection. B6 50 PFU neonates are the same as in Fig 1A and 1B. Data are from 2 similar experiments. (B) 7-day-old B6 neonates were infected with 5 PFU of LCMV Armstrong and treated with 50ug of either anti-CD8 (n = 13), anti-CD4 (n = 16) or IgG control (n = 4) on days 0, 4 and 8 post infection. Data are from 3 similar experiments. (C) Viral load in TCRβ KO neonates was determined by plaque assay in kidney, lung and liver on day 9 post infection with 50PFU of LCMV Armstrong. Data are from two similar experiments. (D) Mortality curve of IgH^a.gld mice treated with anti-IFNγ and Enbrel (entercept) infected with 5 PFU of LCMV Armstrong. Control mice include B6.IgH^a and GLD.IgH^a mice treated with PBS and control IgG1. n = 6–13 mice/group from at least two similar experiments. (E) Mortality curves for perforin KO neonates compared with wildtype B6 neonates after infection with 5 PFU of LCMV Armstrong. n = 11–16 mice/group. Data are from two similar experiments.

over 2 logs higher than that found in B6 neonates, and 100% of TCRβ KO neonates survived until the end of the experiment at day 14 post infection with no signs of illness (Fig 3C).

To more specifically target the mechanism by which T cells were inducing death, a number of genetically modified mouse models were employed. Neonates lacking FasL (gld) infected with 5 PFU of LCMV showed no difference in the kinetics of death or the development of immunopathology (Fig 3D). Likewise, when FasL deficient neonates were treated with anti-IFNγ and soluble TNF receptor (etanercept) to block cytokine signaling, there were no differences in mortality (Fig 3D), indicating that Fas-FasL, IFNγ and soluble TNF do not play a major role in mortality in neonates infected with LCMV. However, when perforin-deficient neonates were infected with 5 PFU of LCMV, they had 100% survival (Fig 3E). These data indicate that in neonates CD8 T cells and perforin are both mediating severe immunopathology and death. Most likely the perforin is released from LCMV-specific CD8 T cells.

## Neonates have replicating virus and LCMV-specific CD8 T cells in their brains

During these experiments, some neonates developed paralysis and seizures prior to death. These findings are not found in adult mice after LCMV Armstrong infection i.p. on day 3 and 5 post infection, low levels of virus were found in the brains of neonates. By day 6 post infection, viral load had expanded to 5.0 $\log_{10}$ PFU/gram of organ, which persisted until day 9 (Fig 4A). By day 14 post infection, all surviving neonates had cleared virus from the brain (Fig 4A). LCMV Armstrong infection in adult mice (i.p.) does not normally gain access to the mature brain, and we were unable to detect virus from the brains of adult mice at day 3, 6 or 9 post infection. In neonates, the peak of viral load in the brain, day 6–9 post infection, correlated with the window of time when neonates were dying. Since CD8 T cells were mediating the mortality of neonates infected with LCMV, we questioned if T cells were infiltrating the brain to attack virus-infected cells. LCMV-specific, IFNγ-producing CD8 T cells were found in the brain of neonates on day 9 post infection (Fig 4B and 4C). These data suggest that the brain was infected in neonates, leading to the recruitment of CD8 T cells, which played a role in the increased mortality.

## Neonates have delayed CD8 T cell responses

The overall CD8 T cell population, which is very low by both percentage and number at the time (day 7 of age) that the neonatal mice were infected with LCMV, was found to be mediating increased mortality. We questioned what was different about the CD8 T cell response or the overall immune response in neonates, to allow for delayed viral clearance and dissemination of virus into the brain. We first examined the kinetics of LCMV-specific CD8 T cell expansion and overall size of the CD8 T cell response.

Neonates infected with low dose LCMV had a reduced LCMV-specific CD8 T cell response as determined by both frequency and total number (Fig 5A and 5C). At day 9 post infection, when the majority of neonates were succumbing to infection, the LCMV-specific response was still relatively low at 9%, compared to 32% in adults. It was not until day 14 post infection (in the few surviving neonates) that the LCMV-specific CD8 T cell response had expanded to similar frequencies as adult controls. The frequency of the total CD8 T response was also lower in neonates than adults, at 2% vs 32% and 12% vs 30%, at days 9 and 14, respectively. The ability for neonatal T cells to expand in response to an LCMV infection is stunted compared to adults. The total number of LCMV-specific CD8 T cells in adult mice expanded 99-fold between day 6 and 9 post infection, while neonatal CD8 T cells only expanded 1.6-fold over this same time period. However, between day 9 and 14 the surviving neonates showed a 40-fold expansion in

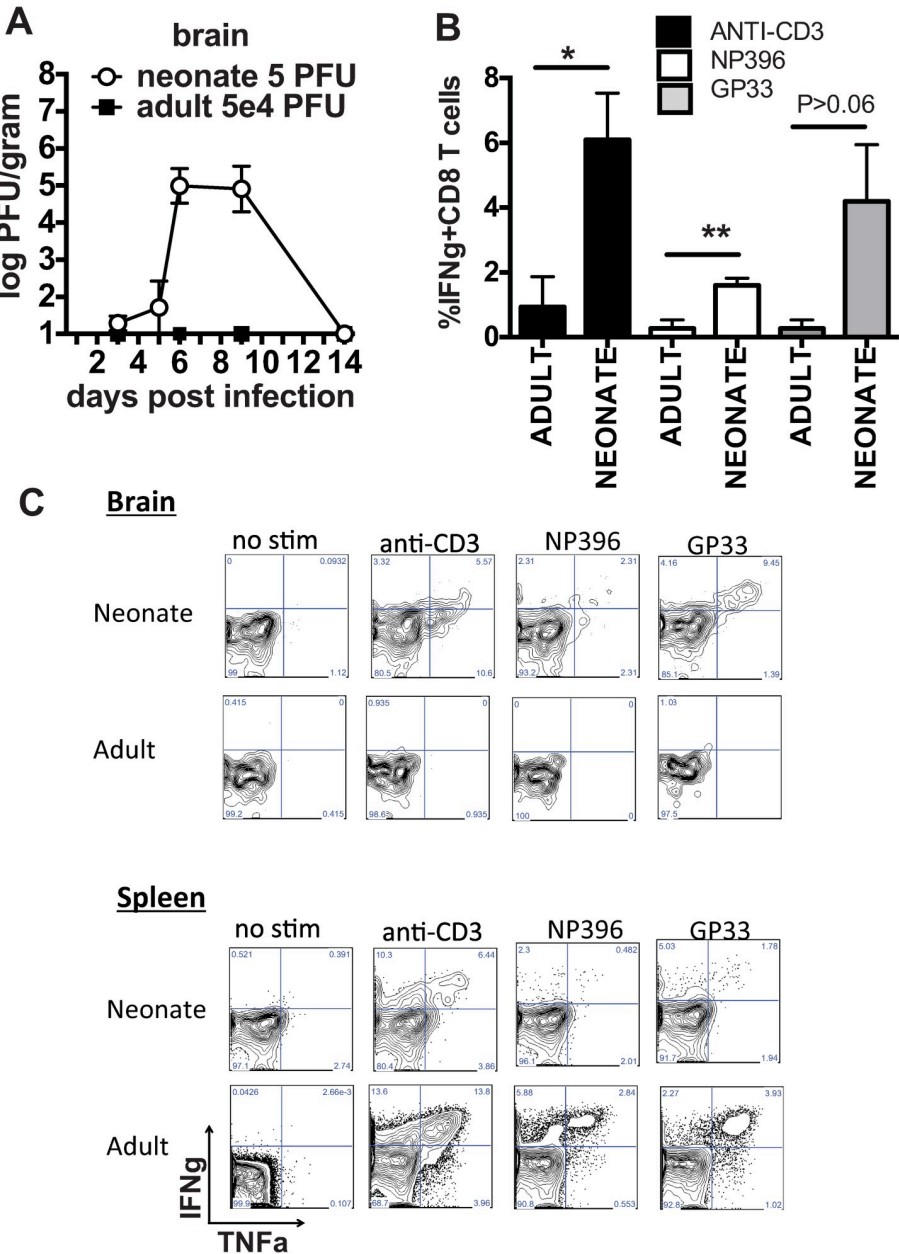

**Fig 4. Neonates infected with LCMV Armstrong have viral dissemination and T cell infiltration into the brain.**
(A) The kinetics of viral replication in the brain of neonates infected with 5 PFU of LCMV and adults infected with $5x10^4$ PFU was determined by plaque assay and converted to $\log_{10}$ PFU/gram of organ. The detection limit of the plaque assay for brain is 1 $\log_{10}$. n = 3–7 mice/group. Data are pooled from at least 2 separate experiments. (B) Lymphocytes were isolated from the brain on day 9 post LCMV Armstrong infection from neonates infected with 5 PFU or adults infected with $5x10^4$ PFU and the frequency of IFNγ producing CD8 T cells was determined after restimulation with LCMV peptides. Data are averages of n = 4–5 mice/group from two similar experiments. (C) Representative FACS plots of IFNγ vs TNF staining gated on CD8+CD44hi T cells from lymphocytes isolated from the brain of neonates and adult mice on day 9 post infection with LCMV.

their LCMV-specific CD8 T cell pool and were no longer significantly reduced compared to adults. These data show that there was a smaller virus-specific T cell pool and a delay in the peak of the LCMV-specific CD8 T cell response in the limited number of surviving neonates

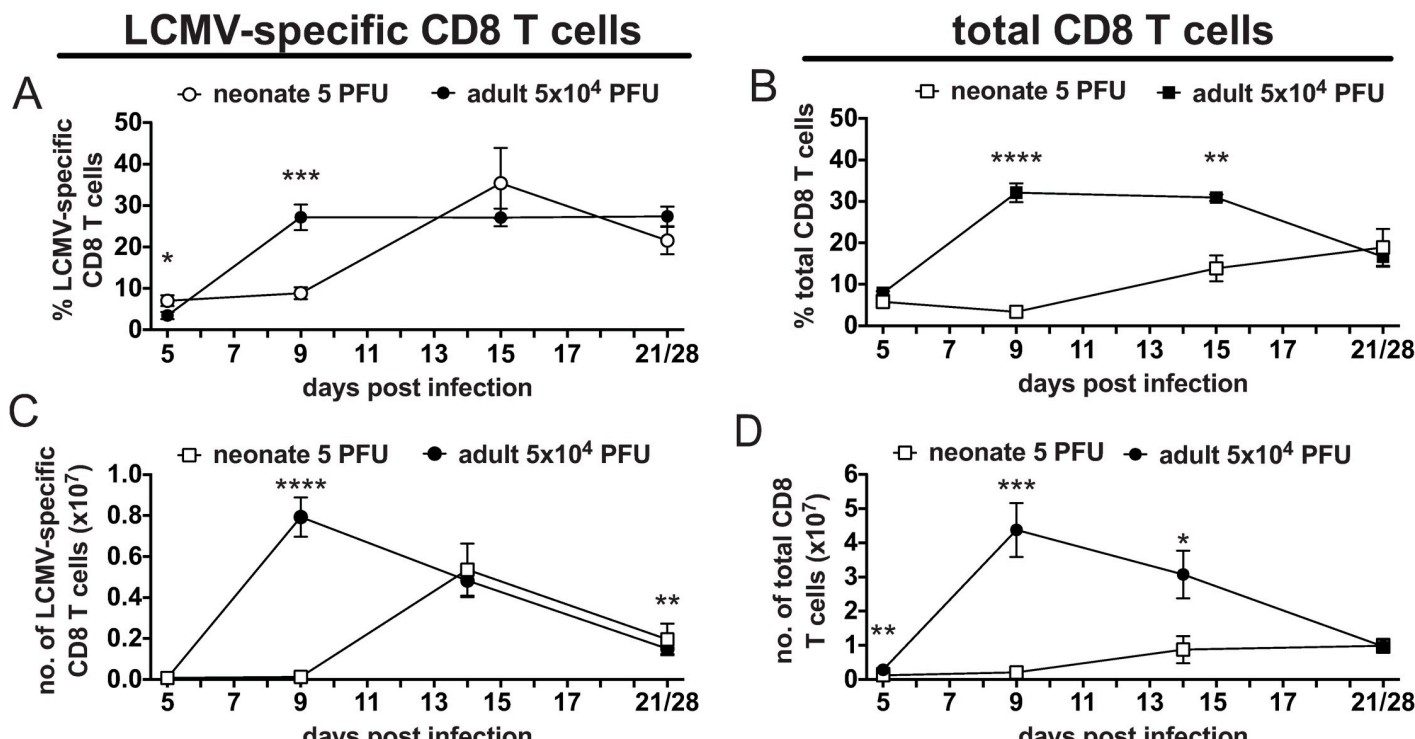

**Fig 5. The peak of the LCMV Armstrong-specific CD8 T cell response is delayed and reduced in neonatal mice.** The kinetics of the total LCMV-specific (A and C) and total (B and D) CD8 T cell response was determine for neonates infected with 5 PFU of LCMV Armstrong compared to adults infected with $5x10^4$ PFU by both frequency (A and B) and total number (C and D). N = 3–15 mice/group. Data are from at least two similar experiments. The total LCMV frequency and number was calculated by summing the individual responses to LCMV epitopes NP396, GP33, GP276 and NP205 based on ICS assay results.

after LCMV infection due to a reduced expansion of CD8 T cells. However, the neonatal mice that survived, cleared the virus essentially by day 14 (Fig 2) and their T cell populations continued to increase as the mouse matures and they developed normal memory CD8 T cell frequencies (Fig 1A–1D), and in that process develop equivalent LCMV-specific CD8 T cell responses to adults (Fig 5).

## Neonatal CD8 T cell responses have a partial exhaustion phenotype

In adult mice infected with an intermediate dose of LCMV clone 13, the T cell-mediated mortality was associated with high antigen loads driving a partial loss of function of the virus-specific CD8 T cells, or partial clonal exhaustion [17–19]. During partial exhaustion, the suboptimally functioning T cells contribute to sustaining viral load, which further drives the immune response and results in lethal immunopathology. In the current study, the high antigen loads and the lack of T cell expansion between day 6–9 post LCMV infection suggest antigen-driven exhaustion could be occurring in neonates. We questioned if neonatal CD8 T cells showed markers of exhaustion.

PD-1 is both an activation marker and an exhaustion maker on antigen-specific CD8 T cells after LCMV infection. At day 9 post infection, the percentage of LCMV epitope-specific cells in the spleen, that were PD-1+ was significantly higher in neonates than adult controls for NP396, GP33, GP276, NP205 and anti-CD3 stimulation (Fig 6A). CD8 T cell exhaustion occurs with a stepwise loss of effector CD8 T cell function. As the ability to produce TNF is lost before IFNγ a ratio of the frequency of TNF to IFNγ producing CD8 T cells can be used to

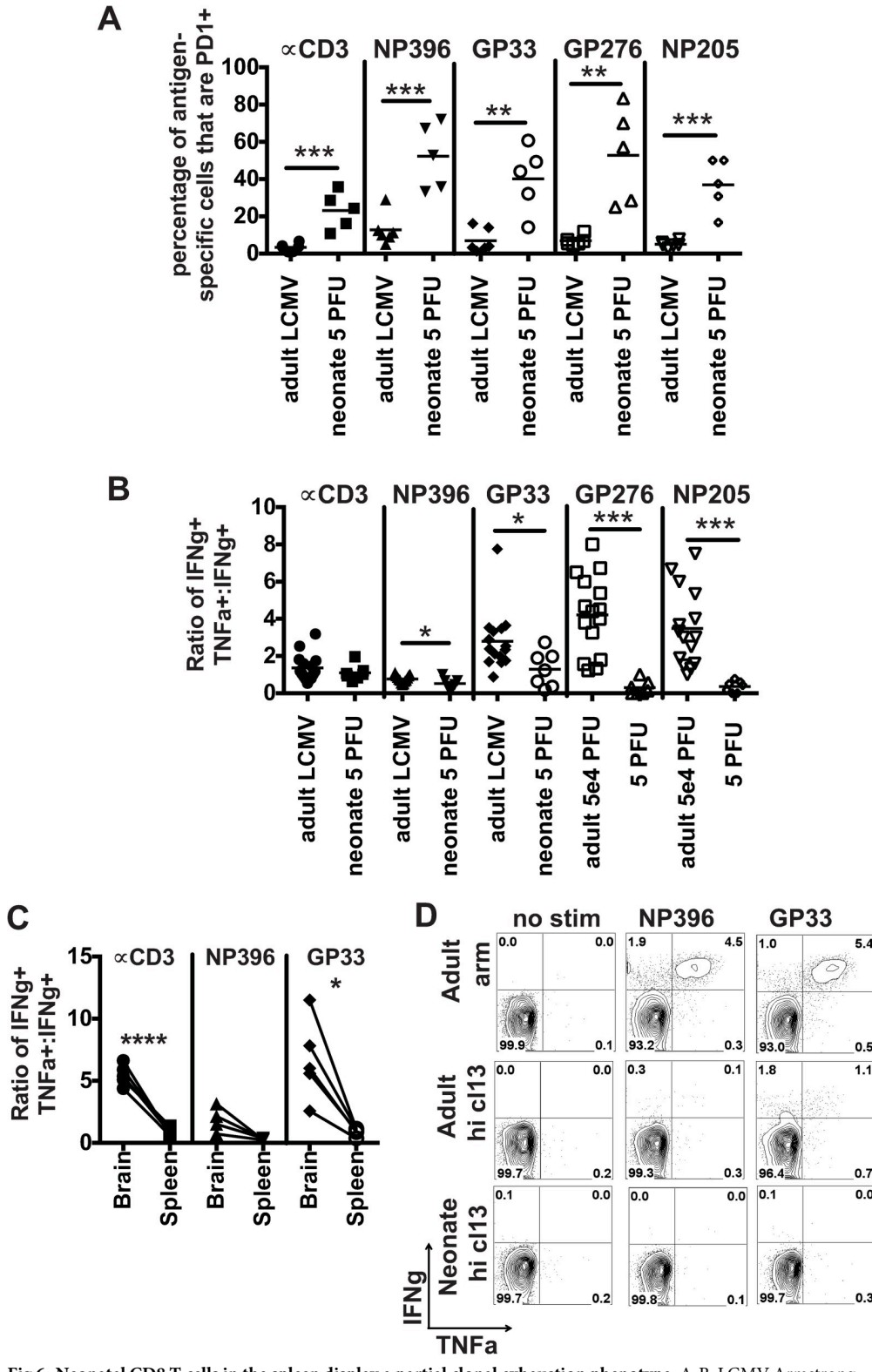

**Fig 6. Neonatal CD8 T cells in the spleen display a partial clonal exhaustion phenotype.** A-B. LCMV Armstrong-specific CD8 T cells from the spleens of neonatal mice (50 PFU) and adult mice (5x10⁴ PFU) were examined on day 9 post infection for PD-1 expression (A) and TNF production (ratio of IFNγ+TNF+ cells to IFNγ-only+ cells) (B). (C) The ratio of IFNγ+TNF+:IFNγ producing cells was examined in the brain and spleen of the same neonate. (D) To determine the ability of neonatal CD8 T cells to undergo complete clonal exhaustion neonates were infected with 2x10⁶

PFU of LCMV clone 13 ip and compared to adult mice infected with either 5x10$^4$ PFU LCMV Armstrong ip or high dose 2x10$^6$ PFU LCMV clone 13 iv. On day 14 post infection, IFNγ and TNF production was examined in splenocytes after peptide stimulation. Data are representative of two similar experiments.

determine the cytokine production capacity of effector T cells [17]. The ratio of IFNγ+TNF + cells to IFNγ-only producing CD8 T cells was lower in neonates after restimulation in vitro with NP396, GP33, GP276 and NP205 (Fig 6B). Taken together these data suggest that partial clonal exhaustion may be one factor that plays a role in the delayed expansion of LCMV-specific CD8 T cells, and this, in turn, may allow for high viral loads and dissemination of virus into the brain.

The partial clonal exhaustion phenotype of neonatal CD8 T cells may seem to be contrary to the finding that neonates are succumbing to perforin-dependent death, as loss of cytolytic ability is the first function lost in the process of clonal exhaustion. If LCMV-specific CD8 T cells undergo partial clonal exhaustion, which involves a loss of perforin, how are CD8 T cells mediating death? We compared the functionality of T cells in the brain versus the spleen and found that within the same neonatal mouse both total CD8 T cells and LCMV GP33-specific CD8 T cells were more functional in the brain as compared to the spleen (Fig 6C). These data would suggest that the CD8 T cells in the brain may be more protected from exhaustion than in the spleen, thus being more functional and able to cause mortality.

As complete functional clonal exhaustion can be a mechanism for protection from T cell-mediated immunopathology [17–19], we questioned if neonates were unable to completely clonally exhaust. Neonates were infected with a high dose (2x10$^6$ PFU) of LCMV clone 13 i.p. All neonates survived until the end of the experiment at day 14 post infection, and the CD8 T cell response underwent complete functional clonal exhaustion by loss of IFNγ and TNF production (Fig 6D). These data suggest that at very high infecting doses with a virus, such as LCMV clone 13, known to induce clonal exhaustion, neonates can exhaust.

## Reduction or elimination of early high viral loads results in complete survival in neonates

Two other related neonate infection models were examined to determine how reducing the early viral load may influence both the T cell response and overall survival. 1. passive immunity from LCMV-immune mothers and 2. infection with a related less virulent arenavirus, Pichinde virus (PICV).

## Maternal antibody protects neonates from death by mediating faster viral clearance

To understand how early control of viral load may impact survival and dissemination of virus into the brain, we utilized a maternal antibody model. Neonates from LCMV-immune mothers infected with 50 or 500 PFU of LCMV Armstrong were completely protected from death (Fig 7A) and had minimal lung and liver infiltrates (Fig 7B). Neonates from LCMV-immune mothers also cleared virus faster and had lower viral loads (Fig 7C). Neonates from LCMV-immune mothers cleared virus from the kidneys by day 9 post infection, whereas surviving neonates from naïve mothers still had ~3 log$_{10}$ PFU/mL at day 14 post infection (Fig 7C). Furthermore, virus was not found in the brains of neonates from LCMV-immune mothers at day 7 post infection. These data suggest that passive immunity could control viral replication early in infection, possibly compensating for the weak neonatal innate immune system, allowing the neonatal CD8 T cell response to control infection without inducing massive collateral damage.

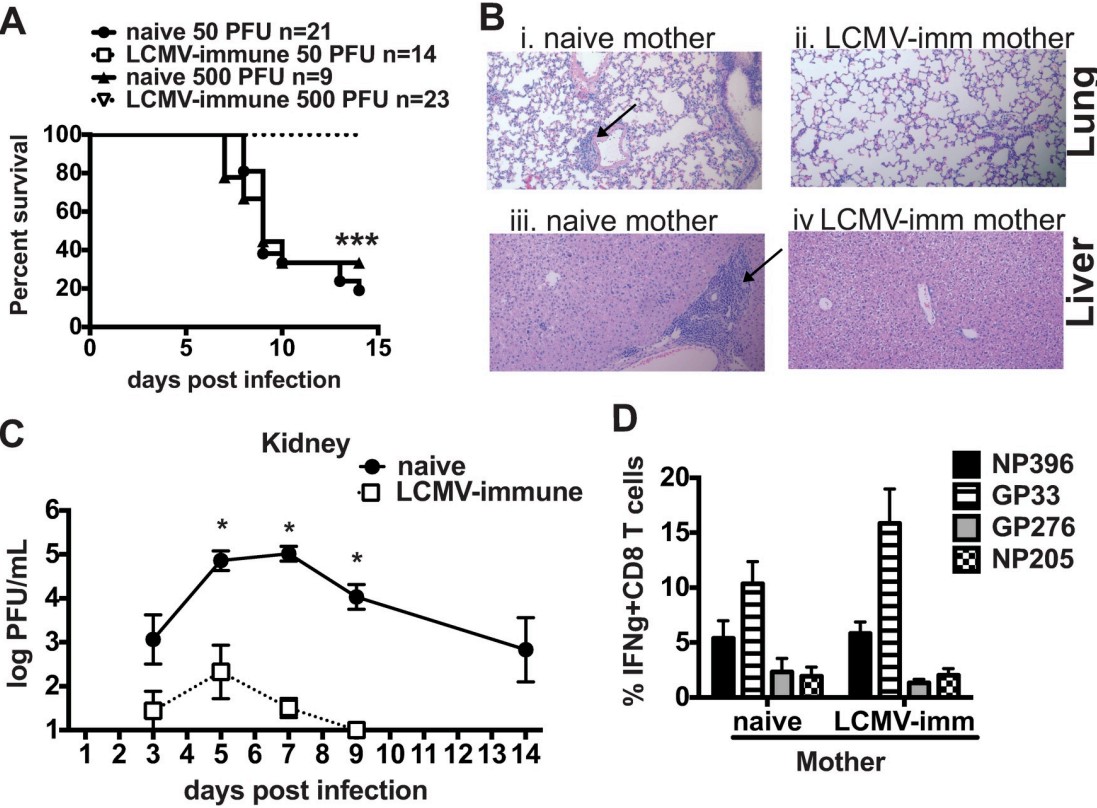

**Fig 7. Passive immunity from LCMV Armstrong-immune mothers protects neonatal mice from LCMV Armstrong-mediated mortality and immunopathology.** (A) Mortality curves from neonates infected with either 50 or 500 PFU of LCMV Armstrong from either naïve or LCMV-Armstrong immune mothers. n = 9–23 mice/group and data are from at least 2 similar experiments. (B) Lung and liver sections were stained with H and E. (Bi) Lung from LCMV Armstrong-infected neonate from a naïve mother showed interstitial mononuclear infiltrates and lymphocyte cuffing around vesicles. (Bii) Lung from infected neonate from LCMV Armstrong-immune mother showed minimal interstitial infiltrates. (Biii) Liver from LCMV Armstrong-infected neonate from naïve mother showed several small pockets of mononuclear infiltrates and a large area of mononuclear lymphocyte cuffing around vessel. (Biv) Liver from LCMV Armstrong-infected neonate from LCMV Armstrong-immune mother showed only minimal areas of mononuclear infiltrates. Arrows highlight pathology. (C) Viral load was determined in the kidney of neonates infected with 50 PFU of LCMV Armstrong from either naïve or LCMV-immune mothers. Viral load was determined by plaque assay and $\log_{10}$ PFU/mL was calculated. The detection limit of the plaque assay for kidney is 1 $\log_{10}$. n = 2–5 mice/group from at least two separate experiments. (D) Immunodominance hierarchies from LCMV Armstrong-infected neonates from either naïve or LCMV Armstrong-immune mothers determined on day 14 post infection by IFNγ production after peptide restimulation of splenocytes. n = 3–13 mice/group from two separate experiments.

One concern with the presence of maternal antibody during vaccination has been that it may clear virus too rapidly, thereby blocking the development of the neonate's own immune response. CD8 T cell responses in neonates from LCMV-immune mothers had similar immunodominance hierarchies and frequencies as to compared with surviving neonates from naïve mothers (Fig 7D). These data indicate that maternal antibody did not block T cell priming, activation and expansion, and it lowered viral load enough to enhance survival.

## Neonatal mice can produce fully functional CD8 T cell responses with similar kinetics to adult mice after PICV infection

To determine if the inability to effectively control LCMV infection was common to other virus infections in the 7-day-old neonate we infected neonates with a related arenavirus, Pichinde virus (PICV). No mortality was found when neonates were infected with either a low $2 \times 10^4$

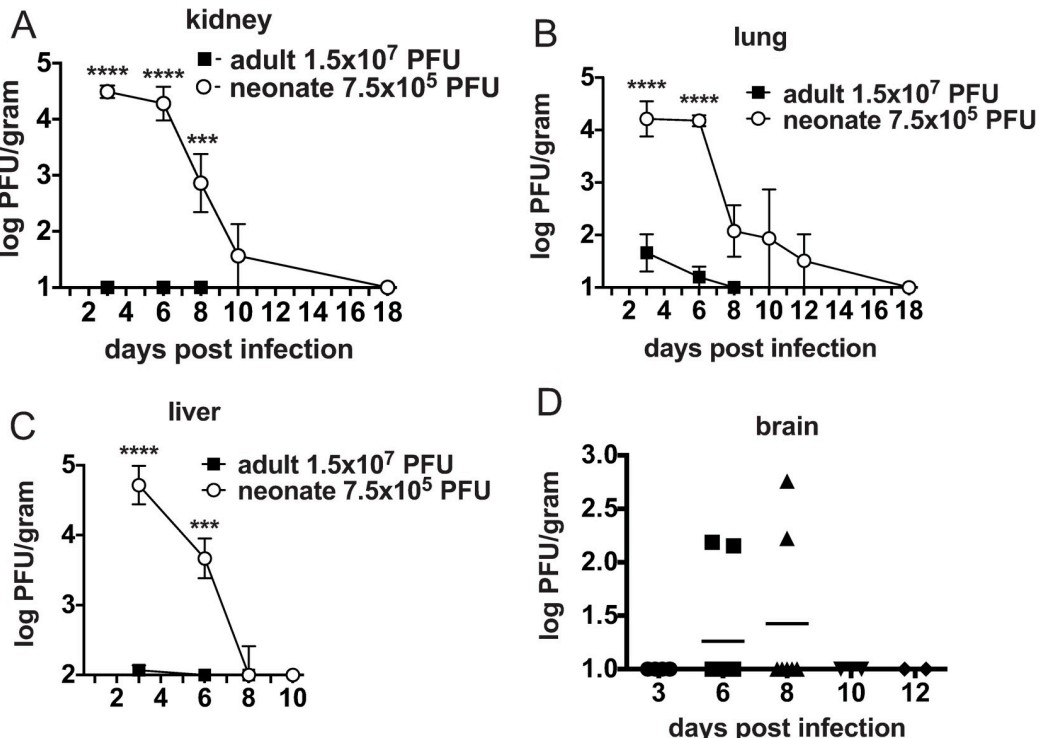

**Fig 8. Neonates infected with a related arenavirus, PICV, have delayed clearance of PICV.** A-D. Viral load was determined in (**A**) kidney, (**B**) lung and (**C**) liver and (**D**) brain of both neonates infected with $7.5x10^5$ PFU and adults infected with $1.5x10^7$ PFU of PICV. Viral load was determined by plaque assay and $log_{10}$ PFU were calculated per gram of organ to compare adult to neonate. The detection limit of the plaque assay for kidney, lung and brain is 1 $log_{10}$; the liver is 2 $log_{10}$. Averages of 3–7 mice/group are shown from at least two separate experiments.

PFU or high $7.5x10^6$ PFU dose of PICV. Compared to adult mice infected with the normally used dose, $1.5x10^7$ PFU, neonates had 347- and 398-fold higher viral loads, which persisted 2 and 10 days longer, respectively, in the lung and liver. While PICV does not replicate in the kidneys of adult mice, in neonates PICV replicated to over 4 $log_{10}$ PFU/gram of organ (Fig 8A–8C).

Compared to LCMV-infected neonates, PICV-infected neonates were able to efficiently control viral replication. In contrast to LCMV infection, where viral loads peaked between days 6–9, PICV viral loads in neonates peaked at day 3, after which PICV titers decreased from kidney, lung and liver and were cleared by day 18 (Fig 8A–8C). Furthermore, peak viral loads in PICV-infected neonates were 20-250-fold lower compared to the same organs of LCMV-infected neonates. Due to the lower virulence of PICV, it was administered at a much higher dose than LCMV ($7.5x10^5$ PFU PICV vs 5 PFU LCMV). The ratio of infecting viral dose and the peak viral load found in neonates was dramatically different between LCMV and PICV. After infection of neonates with $7.5x10^5$ PFU of PICV over 4 $log_{10}$ PFU/gram of organ was detected on day 3 post infection (ratio $10^4$ PFU: $7.5x10^5$ PFU = 0.0133), while after 5 PFU of LCMV over 6 $log_{10}$ PFU/gram of virus was detected (ratio $10^6$ PFU: 5 PFU = 200,000). These data show that neonates were capable of controlling PICV replication early without immuno-pathology even at higher doses.

After LCMV infection, delayed dissemination of virus to the brain was associated with IFNγ and TNF producing CD8 T cells within the brain (Fig 4). We questioned if PICV would also disseminate into the brain even though there was better control of viral replication in the

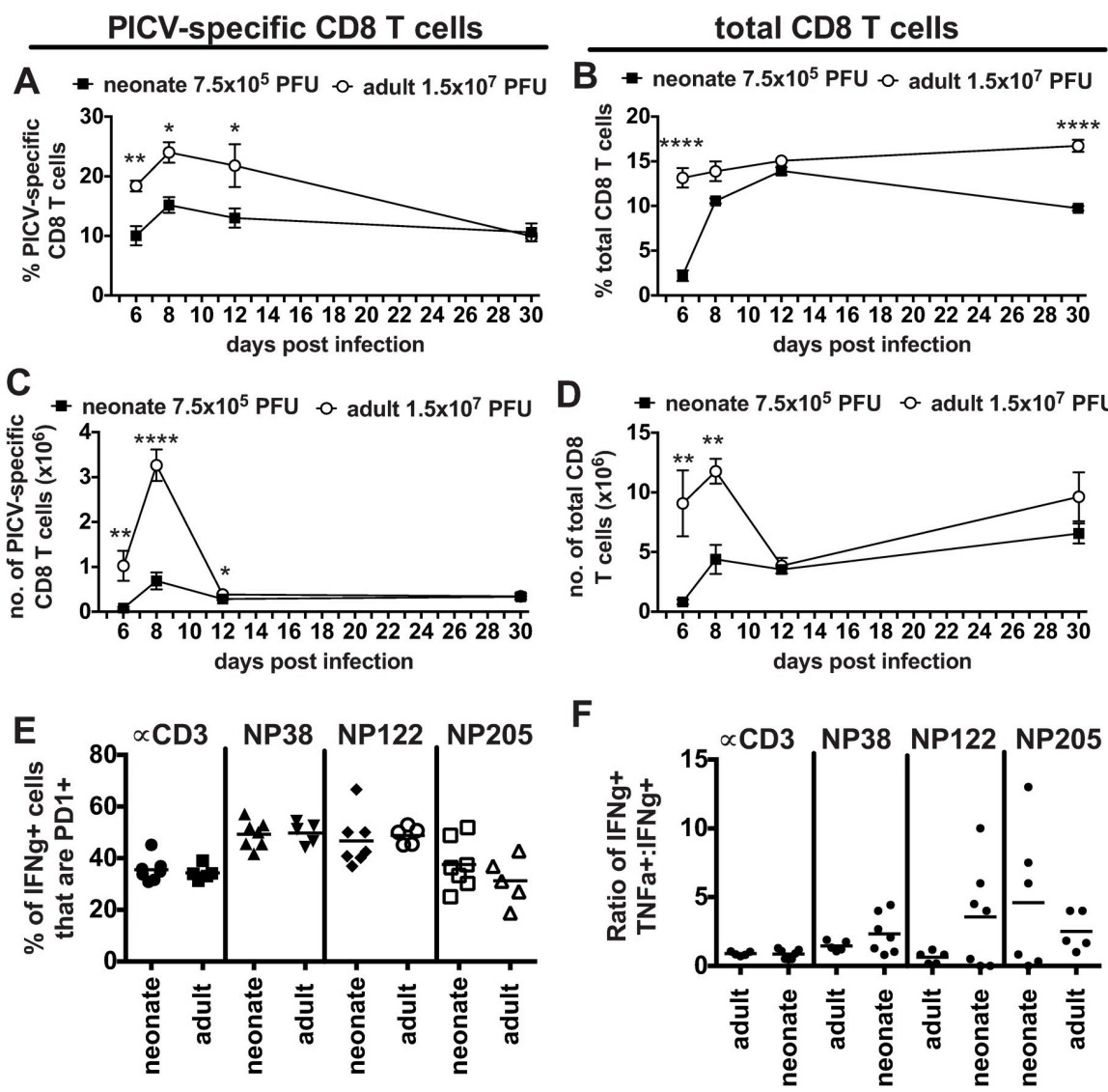

**Fig 9. Neonates infected with a related arenavirus, PICV, have adult-like CD8 T cell responses.** The kinetic of the PICV-specific CD8 T cell response (A and C) and total CD8 T cell response (B and D) in the spleen was determine for neonates infected with $7.5 \times 10^5$ PFU of PICV compared to adults infected with $1.5 \times 10^7$ PFU by both frequency (A and B) and total number (C and D). N = 3–6 mice/ group. Data are from at least two similar experiments. PICV-specific CD8 T cells from spleens of neonatal mice ($7.5 \times 10^5$ PFU) and adult mice ($1.5 \times 10^7$ PFU) were examined on day 8 post infection for PD-1 expression (E) and TNF production (ratio of IFNγ+TNF+ cells to IFNγ-only+ cells) (F). Data is representative of 3 similar experiments. The total PICV frequency and number was calculated by summing the individual responses to PICV epitopes NP38, NP122 and NP205 based on ICS assay results.

periphery. In the majority of neonates, virus was not detectable in the brain, but very low levels of virus (3–4 plaques in the neat dilution) were detected in 2 out of 9 neonates at day 6 and 2 out of 7 neonates at day 8 (Fig 8D).

PICV-infected neonates had similar kinetics in their PICV CD8 T cell responses, with no delay in the peak response when compared to adults (Fig 9A and 9C). However, the frequency and number of PICV-specific T cells were significantly lower in neonatal mice at days 6, 8 and 12 post infection compared to adults, helping to explain why viral loads are higher in neonates (Fig 8A and 8C). The percentage and number of total CD8 T cells were also significantly lower in neonates at day 6 post infection, but by day 12 neonates seemed to "catch up" with adult T

cell responses (Fig 9B and 9D). There also was no evidence of CD8 T cell exhaustion based on TNF+INFg+:IFNg ratio or PD1 expression (Fig 9E and 9F).

## Discussion

Here we show data that is consistent with the concept that the balance between virus load and efficiency of the CD8 T cell response plays an important role in whether neonates survive infection. Fast replicating, low dose LCMV Armstrong infection kills neonates as virus wins over the immune response. The poor early control of viral loads led to partial clonal exhaustion, viral entry into the immature brain and lethal LCMV-specific CD8 T cell infiltration leading to death. If the poor early viral control was compensated for either by maternal antibody or infection with a less virulent arenavirus, PICV, the immune system wins over the viruses. Overall, this suggests that neonatal CD8 T cells are fully functional although present at lower numbers. These data are important in understanding how to best vaccinate and protect newborns, which are highly sensitive to viral infections.

In adults, there is a similar balance between the ability of the immune system to control viral load and the efficiency of the T cell response to determine the severity of disease after LCMV infection. In adult mice infection with LCMV Armstrong i.p. and high dose LCMV clone 13 i.v. both result in minimal immunopathology, but differ significantly in their viral loads and T cell efficiency. Adult mice infected with LCMV Armstrong quickly control virus with a fully functional immune response, while after high dose LCMV clone 13 infection high antigen loads induce complete clonal exhaustion and T cells do not have the capability to cause immunopathology, even though virus persists [17–19]. When a mediocre cytotoxic T cell response is present in a host with high systemic viral loads, immunopathology can develop, as found when adult mice are infected with an intermediate dose of LCMV clone 13 i.v. Only partial clonal exhaustion occurs, leaving a less efficient immune response that induces severe immunopathology in response to systemic viral infection [17–19].

In the present study, neonatal mice infected with LCMV Armstrong, even at very low doses (5 PFU), resulted in severe immunopathology and 80% mortality. Similar to the intermediate dose of LCMV clone 13 in adult mice, mortality in neonates was CD8 T cell-mediated. In both infection models, loss of CD8 T cells either using depletion antibodies or genetic modification, resulted in less severe immunopathology and reduced mortality [17]. The role of T cells in immunopathology is also found in other human pathogens, such as Lassa virus. Lassa virus is a related arenavirus and the causative agent for Lassa fever, a hemorrhagic fever that causes several thousand deaths each year. Mice expressing human HLA-A2 had high levels of viral replication and liver immunopathology after Lassa virus infection that was reduced after depletion of CD4 and CD8 T cells [29]. In this study, we used blocking antibodies or genetically modified mice to determine the specific mechanism for T cell-mediated neonatal mortality and found that Fas-FasL interactions, IFNγ or soluble TNF were not required. When adult mice are infected with an intermediate dose of LCMV clone 13 the LCMV-specific CD8 T cell response undergoes partial clonal exhaustion, as shown by an increase in the expression of PD-1 on LCMV-specific T cells and a loss of TNF production, but CD8 T cells are still capable of producing IFNγ [17–19]. In the current study, neonatal LCMV-specific CD8 T cells had a similar partial clonal exhaustion phenotype in the spleen with an increased percentage of PD-1 + cells and a loss of TNF production. This partial clonal exhaustion phenotype was not due to any inherent differences between adults and neonates as after PICV infection there was no difference between neonatal and adult CD8 T cells for either of these parameters.

Cytotoxicity through granzymes and perforin is lost during clonal exhaustion [30], making the perforin-dependent death observed in neonates inconsistent with the partial clonal

exhaustion phenotype of the CD8 T cells in the periphery of the same mice. However, within the brains of neonates the LCMV-specific T cells did not exhibit the same partial clonal exhaustion phenotype, indicating that these cells were cytotoxic, and within the brain these cells could be a major player in lethality. Interestingly, the brain may be a sight where T cells are protected from clonal exhaustion. At this time, we do not know if there is anything immunologically different about the brain environment that would allow the CD8 T cells to survive in the brain and not in the periphery. It is possible that the CD8 T cells may be protected from exhaustion due to the delayed kinetics of high viral loads in the brain compared to the periphery where CD8 T cells were partially exhaust. It may just be a matter of the kinetics of the virus which replicates at high levels in the kidney, liver and lung as early at day 2 of infection but does not reach significant levels in the brain until day 6 (because of the blood brain barrier). This may give enough time for the CD8 T cells that have come from the peripheral circulation to become fully functional before the viral load in the brain starts to increase at day 6 when the peripheral CD8 T cells are exhausted and losing control of the virus.

To shift the balance between T cell or immune system efficiency and viral load, passive immunity from LCMV-immune mothers was utilized. In a previous study, Balb/c LCMV-immune nursing mothers could protect 100% of 10 day old pups from 1,000 PFU of LCMV Armstrong (strain 4) injected intracranial [31]. Maternal antibody from LCMV Armstrong (strain 4) immunized mothers had a measurable effect on decreasing viral load and blocked the development of persistent LCMV infection in neonatal mice less than 24 hours old [32]. In that study the maternal antibody protection depended on β2m expression by the neonates and the neonates need T cells. Although the virus-specific CD8 T cells were not quantified, that study did establish a role for CD8 T cells in order for neonates to be passively protected. Furthermore, maternal antibody was found to protect suckling rats from developing LCMV-mediated brain immunopathology [33]. However, none of these studies examined the development of CD8 T cell responses in the neonates in the presence of maternal antibody. In the current study, passive antibody protected-neonates had faster viral clearance, with over a 1000-fold reduction in peak viral titer. These data would suggest that the presence of maternal antibody compensates for the defects in the neonatal immune system and helps keep viral load in check early during infection, thereby allowing for the neonatal T cell response to efficiently clear LCMV. The impact of maternal antibody on the development of a protective immune response in newborns and infants is a concern and impacts vaccination schedules in humans. However, in areas of high infant mortality, earlier vaccination in the presence of maternal antibody followed by booster vaccination improved mortality over delayed vaccination. For example, increased levels of protective antibodies were found when children were vaccinated against measles virus at 4.5 months and boosted at 9 months, compared to children vaccination only at 9 months in low-income countries, where measles virus infections readily occur [34]. This may be a safer way to give the live-virus vaccines to the developing immune systems of young children, especially in areas where significant virus-mediated death is prevalent.

The use of a less virulent arenavirus, PICV, also resulted in 100% survival of neonates due to lower viral loads that did not overpower the immune response or lead to clonal exhaustion but allowed for an efficient immune response to develop. PICV-infection was well tolerated, as neonates infected with up to $7.5 \times 10^5$ PFU had no mortality, while neonates infected with only 5 PFU of LCMV had up to 80% mortality. In LCMV-infected neonates, virus replicated to higher levels, and the peak of viral load was delayed to day 9, while virus load peaks at day 2 after PICV-infection, similar to adults.

The mortality found in this study is unique to the age of infection and illustrates the altered dynamics of the neonatal immune system compared to that of adults. Newborn mice infected with LCMV within the first 24 hours of life become tolerized to LCMV antigen and do not die,

but become persistently infected [22, 23]. In the current study, by day 12 of age, mice could control infection and did not die from immune-mediated pathology. The day 7 neonate represents a vulnerable window, where mice are capable of producing strong cytotoxic T cell responses but depending on the speed that the virus replicates the mice can either quickly clear infection, as they do after PICV infection, or develop severe immunopathology, as occurs after LCMV infection.

Previous studies have shown that the infection conditions determine the type of immune response which develops in neonates; low, persistent doses of antigen can promote CTL/Th1 responses, while higher doses will induce Th2 or tolerance [7]. 1000 PFU of murine leukemia virus induced a IL-4 producing, Th2-skewed response in 2 day old mice, while 1 PFU resulted in IFNγ-producing cytotoxic CD8 and Th1-skewed CD4 T cell responses [12]. 7-day-old neonatal mice have IFNγ producing CD8 T cells after infection with RSV [35], attenuated Vaccinia virus [36] and attenuated Listeria [37]. CD8 T cells from 7-day-old neonatal mice proliferate and differentiate into effector cells more rapidly than adult CD8 T cells when they are both primed within the same host. In an adoptive transfer model, HSV-specific transgenic CD8 T cells isolated from adult or 7 day old neonatal mice were compared after infection within the same host [8]. The rapid proliferation of neonatal cells in this model led to terminally differentiated effectors and few memory cells within the neonatal cell population, compared to the heterogeneous effector-memory population of adult cells. These findings would suggest that due to the reduced number of immune cells in the neonate it is necessary for them to undergo rapid expansion in response to infection. Unfortunately, this results in little memory formation, but is necessary for the neonate to survive infection. In the current study, it was unexpected that neonates would undergo T cell-mediated death after LCMV infection as T cells are present at such low numbers at day 7 of age. However, the ability of neonatal T cells to rapidly expand and proliferate can compensate for the low numbers. After PICV infection, neonates could produce responses similar to adults in terms of T cell functionality, even though they are reduced in frequency and total number. After LCMV-infection, on the other hand, the virus replicated faster and overwhelmed the neonatal immune system leading to partial clonal exhaustion, virus entry into the brain and eventually death.

Understanding how the immune system develops and how different ages have different susceptibilities can provide vital information that can dictate how we immunize children. Newborns are Th2-prone, but this study shows that it is infection-dependent. Studies in low-income countries where infant mortality is a major problem have shown that Th1 skewing vaccines, such as measles and BCG, non-specifically protect children from death from infections besides measles virus and tuberculosis [38–41]. This suggests that immunizations can alter or train the immune system. Additionally, this study further promotes maternal immunization as a way to protect newborns from fast-replicating viral infections and may assist in live virus vaccinations.

## Materials and methods

### Ethics statement

This study was done in compliance within the guidelines of our Institutional Animal Care and Use Committee (Protocol A-305) following international guidelines for ethical treatment of animals.

### Mice

C57BL/6J (B6, H-2b), Perforin KO, TCRβKO and IgHᵃ GLD (FasL deficient) mice were purchased from The Jackson Laboratory (Bar Harbor, ME). 5–6 week of age males were bred with

females at 8–10 weeks of age. Neonates were infected at day 7 of age. All mice were maintained under specific pathogen free conditions at the Department of Animal Medicine, University of Massachusetts Medical School.

## Viruses

LCMV (Armstrong strain) and PICV (AN3739 strain) stocks were propagated in BHK21 baby hamster kidney cells [42]. To control for culture contaminants supernatants from PICV-infected BHK cells were purified through a sucrose gradient and diluted in Hank's balanced salt solution (HBSS), and LCMV was diluted >40-fold in HBSS.

## Infections

Adult mice 6–8 weeks old were immunized intraperitoneally (i.p.) with $5x10^4$ plaque-forming units (PFU) of LCMV Armstrong, $2x10^7$ PFU of PICV or $2x10^6$ PFU of LCMV clone 13 intravenously (i.v.). Neonatal mice were infected with 5, 50 or 500 PFU of LCMV Armstrong, $2x10^6$ PFU of LCMV clone 13, or $2x10^4$ or $7.5x10^6$ PFU of PICV in 50 μl ip.

## Peptides

LCMV-specific peptides: NP396-404 (FQPQNGQFI; Db), GP33-41(KAVYNFATC; Db), GP276-286 (SGVENPGGYCL; Db) and NP205-212 (YTVKYPNL; Kb). PICV-specific peptides NP38-45 (SALKFHKV; Kb), NP122-132 (VYEGNLTNTQL; Db) NP205-212 (YTV KFPNM; Kb). Peptides were at 90% purity by reverse phase-HPLC from 21st Century Biochemicals (Marlboro, MA).

## CD4 and CD8 depletion

Neonates were injected with 50μg in 50μl of anti-CD4 (clone GK1.5) anti-CD8 (clone 2.43) antibodies or IgG control LFT2 on days 0, 4 and 8 of LCMV infection.

## Viral titer

LCMV and PICV viral titers were determined in spleens, kidneys, brains or lung by plaque assay with serial dilutions of 10% tissue homogenate or serum on American Type Culture Collection Vero cells [43]. Plaques were counted at day 4 and 5 for PICV and day 5 and 6 for LCMV. Viral loads are shown as $log_{10}$ PFU/gram of organ when comparing adult viral load to neonates and $log_{10}$/mL when comparing one group of neonates to another.

## Lymphocyte preparation and intracellular cytokine staining

Mice were perfused with 10mL of HBSS prior to isolation of brains to remove contaminating peripheral blood. Brains of mice were processed through mesh similar to spleens, spun down and resuspended in 30% percoll in RPMI. Samples were then under laid with a 70% percoll in RPMI and spun at 500 g for 30 minutes at 18˚C with no deceleration brake. Lymphocytes were collected from the 30/70 interface. Single cell suspensions from spleen were treated with 0.84% $NH_4Cl$ to lyse red blood cells. Single cell suspensions were stimulated with 1uM of peptide or 5 μg/ml anti-CD3 (145-2C11) and incubated with Golgi plug (BD Bioscience) and recombinant human IL-2 for 4.5 hours at 37˚C. Samples were treated with FC receptor blocking antibody and stained for CD8 and IFNγ (BD Bioscience clone XMG1.2). Samples were collected on a LSRII (BD Bioscience) and analyzed with FlowJo software (Tree Star Inc.).

## Histology

Lungs, livers and fat pads from mice were collected, fixed in 10% neutral buffered formaldehyde and paraffin-embedded. Tissue sections (5 μm) were stained with hematoxylin and eosin (H&E) and analyzed microscopically by a pathologist.

## Statistics

Descriptive statistics are expressed as mean +/- standard error of the mean. Standard error of the mean is displayed in all figures where relevant unless stated differently in figure legends. Statistical analysis was done using the Student's T test when comparing two groups and ANOVA when comparing three or more (with adjusted p values). Mortality curves were analyzed with log rank test using Prism software (Graphpad software, La Jolla Ca). $^*p<0.05$, $^{**}p<0.01$, $^{***}p<0.001$ $^{****}p<0.0001$.

## Acknowledgments

We would like to acknowledge Carey Zammitti and Keith Daniels and for their technical assistance and Ray Welsh for discussion and critical reading of the manuscript. The contents of this publication are solely the responsibility of the authors and do not represent the official view of the NIH.

## Author Contributions

**Conceptualization:** Laurie L. Kenney, Liisa K. Selin.

**Data curation:** Laurie L. Kenney, Erik P. Carter.

**Formal analysis:** Laurie L. Kenney, Erik P. Carter.

**Funding acquisition:** Liisa K. Selin.

**Investigation:** Laurie L. Kenney, Erik P. Carter.

**Methodology:** Laurie L. Kenney, Erik P. Carter.

**Project administration:** Liisa K. Selin.

**Supervision:** Liisa K. Selin.

**Visualization:** Laurie L. Kenney, Liisa K. Selin.

**Writing – original draft:** Laurie L. Kenney, Liisa K. Selin.

**Writing – review & editing:** Laurie L. Kenney, Anna Gil, Liisa K. Selin.

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
