## [Decision Letter · Decision Letter 0]

15 Apr 2020

Dear Dr. Selin,

Thank you very much for submitting your manuscript "T cells in the brain enhance neonatal mortality during peripheral LCMV infection" for consideration at PLOS Pathogens. As with all papers reviewed by the journal, your manuscript was reviewed by members of the editorial board and by several independent reviewers. The reviewers appreciated the attention to an important topic. Based on the reviews, we are likely to accept this manuscript for publication, providing that you modify the manuscript according to the review recommendations. At this time, we are not requesting that you do any new experiments (although small-scale experiments that strengthen the paper of course are appreciated)  but that you submit a detailed answer to all of the reviewers comments and that you modify the manuscript to indicate possible caveats, etc. to your study.

Sincerely,

Luis J. Sigal, D.V.M.; Ph.D.

Associate Editor

PLOS Pathogens

Michael Diamond

Section Editor

PLOS Pathogens

Kasturi Haldar

Editor-in-Chief

PLOS Pathogens

orcid.org/0000-0001-5065-158X

Michael Malim

Editor-in-Chief

PLOS Pathogens

orcid.org/0000-0002-7699-2064

Reviewer Comments (if any, and for reference):

Reviewer's Responses to Questions

**Part I - Summary**

Reviewer #1: The outcome to systemic virus infection in adults can vary depending on inoculum dose, dissemination, and resulting T cell responses, which can either control infection or cause pathogenesis. Neonatal mice have immature innate and adaptive immune systems, so it is unknown how infection dose affects outcome to infection. Herein, 1-week old mice were found to be highly susceptible to very low doses (5 PFU) of LCMV-Armstrong, a viral strain that does not cause pathogenesis in adult mice. LCMV-Armstrong was not controlled early after challenge, allowing for dissemination into the brains of neonatal mice. While CD8+ T cells in the spleen failed to express antiviral cytokines, CD8+ T cells in the brain remained proliferative and were protected from exhaustion, leading to perforin-dependent CD8+ T cell-mediated immunopathology in the brain. Neonates from LCMV-immune moms were passively protected and showed lower initial burdens of LCMV-Armstrong and generated strong peripheral CD8+T cell responses. The virus did not disseminate to the brain in the passively protected mice, allowing for survival. Infection of neonates with Pichinde virus that replicates to only low levels failed to cause pathogenesis in neonates.

Overall, the data show that infection dose impacts outcome to infection in neonates; the weakened initial immune response allows for virus dissemination to the brain where CD8+ T cells cause pathogenesis; passive immunity that effectively lowers early virus growth allows neonates to generate protective CD8+ T cell responses. Overall, this is a well-written manuscript describing several very interesting observations though no new mechanistic insights are highlighted.

Reviewer #2: “T cells in the brain enhance neonatal mortality during peripheral LCMV infection” is a manuscript by Kenney and colleagues that seeks to understand the neonatal immune response to viral infection. The authors demonstrate that neonatal mice have limited numbers of CD4+ and CD8+ T cells that increase over time. They then show low dose Armstrong infection is lethal to 7 day old animals, but mice older than 12 days are protected. This effect is observed between 5-500 pfu of the Armstrong strain. Viral titers in these animals are higher than adult animals infected with higher doses. Genetic depletion of T cells rescues the lethality, while singular depletion of either CD4+ or CD8+ T cells does not. Antibody-mediated neutralization of IFNg and TNFa is not sufficient for rescue, but loss of perforin is. The authors then examine immune responses in the brains of surviving animals and find a delay the antigen-specific CD8+ T cell response in neonates. The virus-specific cells in the brains appear to have more PD-1. Two other neonatal models are used: a LCMV-immune maternal model and pichinde virus infection. While these studies are in an important area there are some major problems that preclude publication in the current form. They include:

Reviewer #3: This manuscript examines the outcome of acute LCMV infection in neonatal mice infected at 1 week after birth. Using the acute and normally nonlethal Armstrong strain of LCMV, the authors show that ~80% of neonatal mice infected with as low as 5 PFU of virus succumb to infection between days 8-14 post-infection. Compared to adult mice, the neonatal mice failed to control initial virus replication in the Kidney, Lung and Liver. TCRb-deficient mice were completely protected from LCMV-induced mortality indicating T cells play an important role in mediating the lethal immunopathology. Antibody-mediated depletion of either CD4 or CD8 T cells resulted in some enhanced survival with the greater effect observed following CD8 T cell depletion. Additional studies using perforin-deficient mice demonstrated that complete survival of neonatal mice lacking perforin suggesting a role for perforin-mediated lysis by CD8 T cells in the immunopathology. A wealth of additional data is provided indicating that the CD8 T cells exhibit a partially exhausted phenotype in the neonatal mice and that virus is able to enter the brain in neonatal mice and that CD8 T cells can be isolated from the brains that are more functional than those found in the periphery. Neonatal mice born from LCMV-immune mothers were protected from lethal LCMV-induced disease via the maternal antibodies they received. In addition, neonatal mice infected with a less virulent arenavirus, Pichinde virus, did not exhibit lethal disease. Overall, this manuscript is well written and the authors provide convincing data in support of their primary conclusions.

**Part II – Major Issues: Key Experiments Required for Acceptance**

Reviewer #1: Concerns/comments.

1. Figures 4 and 6 show that neonatal mice have CD8+ T cells in the periphery that are reduced in magnitude and can develop an exhaustion phenotype, whereas T cells in the brain are protected from T cell exhaustion following Armstrong. This seems to be a major finding but is not further explored or explained mechanistically. What is special about the neonatal brain that would allow T cells to escape exhaustion? Do neonates show T cell exhaustion in the brain following LCMV-Clone13 (Figure 6C-D)?

Reviewer #2: 1. It is not novel, even within the limited field of LCMV immunobiology, that the system can have different effects (memory, exhaustion, death) with different strains, doses and backgrounds. The novel finding here is that 7 day old mice undergo immunopathology but the mechanisms are poorly explored. The authors state that cytolysis should be shut off, but no data are provided. They should either test in a Cr release assay, or as a surrogate perform CD107a and b staining and determine if this is altered.

2. Outside of PD-1, there really are no markers of exhaustion analyzed. What about other inhibitory receptors? Levels of Tox?

3. The switching between different virus doses, sometimes even within a figure is extremely confusing. Is there a way to use one dose?

4. Are the cells in surviving neonates at ~60 days memory or exhausted?

5. To this reviewer at least, the comparison of much higher doses in older animals for both LCMV and PiV is meaningless. Isn’t the appropriate comparison older mice with the low doses? Even if the titers are below the detection limit is this more relevant?

6. Conditions for CD3 stimulation of ICS need to be described.

7. How total LCMV-specific response calculated needs to be described.

8. Are Figure 6A and B spleen or brain, it is very difficult to determine.

9. Figure 1G and 7B have confusing labeling. Why not add appropriate tissue to the right of the boxes?

10. Do the mothers in the LCMV immune experiment have neutralizing antibodies? This data should be shown.

11. All viral titer figures need detection limit added.

12. There are a lot of typographical mistakes in the manuscript. In multiple places morality is used instead of mortality. An arrow guiding the reader to pertinent pathology would be helpful.

13. The figure legends 1 and others, refer to “the same day 7 50 pfu group is shown in A and B for comparison”. It is unclear what this qualifier refers to.

14. References 21 and 22 do not demonstrate 3 aa changes between Armstrong and Clone 13!

15. The order of exhaustion described the authors Il-2 then IFNg then TNFa is not correct in Clone 13. See reference 15 for correct order!

Reviewer #3: None

**Part III – Minor Issues: Editorial and Data Presentation Modifications**

Reviewer #1: 2. Figure 8J-K is not described.

3. Discussion. Reference 33 showed that maternal antibody can protect neonates during infection and the protection depended on b2m expression by the neonates and the neonates need T cells. While it is correct that virus-specific CD8+ T cells were not quantified, that paper established a role for CD8+ T cells in order for neonates to be passively protected. The effect of maternal antibody on virus loads was measured in that paper. Please clarify.

4. It would be helpful to indicate in each legend where LCMV-Armstrong was used, rather than just “LCMV”, and specify that T cells from the spleen were analyzed (unless it is otherwise).

5. Figure 5A,5C legend. Which epitopes were assessed ?

6. In reference to Fig5A-C, the text says “….lower than adults, at 89.5% and 55%....”. It would be clearer to refer to the percentages in the plot.

Reviewer #2: (No Response)

Reviewer #3: A few minor points:

1. Third sentence of the last paragraph of the introduction, I believe the authors meat "Mortality" and not "Morality."

2. Were combined CD4 and CD8 T cell depletions performed? If there were done, this would be a nice addition to Fig 3b.

3. Were statistics performed on the survival curves shown in Fig 3A and 3E? If so, these should be added to the figure.

4. Was the function of CD4 T cells assessed? If so, this would be a nice addition to the manuscript.

PLOS authors have the option to publish the peer review history of their article (what does this mean?). If published, this will include your full peer review and any attached files.

Reviewer #1: No

Reviewer #2: No

Reviewer #3: No
---

## [Editor Report · Decision Letter 1]

14 Oct 2020

Dear Dr. Selin,

We are pleased to inform you that your manuscript 'T cells in the brain enhance neonatal mortality during peripheral LCMV infection' has been provisionally accepted for publication in PLOS Pathogens.

Best regards,

Luis J. Sigal, D.V.M.; Ph.D.

Associate Editor

PLOS Pathogens

Michael Diamond

Section Editor

PLOS Pathogens

Kasturi Haldar

Editor-in-Chief

PLOS Pathogens

orcid.org/0000-0001-5065-158X

Michael Malim

Editor-in-Chief

PLOS Pathogens

orcid.org/0000-0002-7699-2064
---

## [Editor Report · Acceptance letter]

7 Dec 2020

Dear Dr. Selin,

We are delighted to inform you that your manuscript, "T cells in the brain enhance neonatal mortality during peripheral LCMV infection," has been formally accepted for publication in PLOS Pathogens.

Best regards,

Kasturi Haldar

Editor-in-Chief

PLOS Pathogens

orcid.org/0000-0001-5065-158X

Michael Malim

Editor-in-Chief

PLOS Pathogens

orcid.org/0000-0002-7699-2064